# COVID-19, Possible Hepatic Pathways and Alcohol Abuse—What Do We Know up to 2023?

**DOI:** 10.3390/ijms25042212

**Published:** 2024-02-12

**Authors:** Agata Michalak, Tomasz Lach, Karolina Szczygieł, Halina Cichoż-Lach

**Affiliations:** 1Department of Gastroenterology with Endoscopy Unit, Medical University of Lublin, Jaczewskiego 8, 20-954 Lublin, Poland; halina.lach@umlub.pl; 2Department of Orthopedics and Traumatology, Medical University of Lublin, Jaczewskiego 8, 20-954 Lublin, Poland; tomasz.lach@umlub.pl; 3Clinical Dietetics Unit, Department of Bioanalytics, Medical University of Lublin, Chodźki 7, 20-093 Lublin, Poland; karolina.szczygiel@umlub.pl

**Keywords:** SARS-CoV-2, COVID-19, alcohol, liver, cirrhosis, alcohol-related liver disease

## Abstract

The pandemic period due to coronavirus disease 2019 (COVID-19) revolutionized all possible areas of global health. Significant consequences were also related to diverse extrapulmonary manifestations of this pathology. The liver was found to be a relatively common organ, beyond the respiratory tract, affected by severe acute respiratory syndrome coronavirus-2 (SARS-CoV-2). Multiple studies revealed the essential role of chronic liver disease (CLD) in the general outcome of coronavirus infection. Present concerns in this field are related to the direct hepatic consequences caused by COVID-19 and pre-existing liver disorders as risk factors for the severe course of the infection. Which mechanism has a key role in this phenomenon—previously existing hepatic disorder or acute liver failure due to SARS-CoV-2—is still not fully clarified. Alcoholic liver disease (ALD) constitutes another not fully elucidated context of coronavirus infection. Should the toxic effects of ethanol or already developed liver cirrhosis and its consequences be perceived as a causative or triggering factor of hepatic impairment in COVID-19 patients? In the face of these discrepancies, we decided to summarize the role of the liver in the whole picture of coronavirus infection, paying special attention to ALD and focusing on the pathological pathways related to COVID-19, ethanol toxicity and liver cirrhosis.

## 1. Introduction

Severe acute respiratory syndrome coronavirus-2 (SARS-CoV-2) has been one of the most significant global threats for almost four years, which has revolutionized medicine and social life—in other words, almost everything [1,2]. Even though the number of coronavirus disease 2019 (COVID-19) cases has exceeded 773 million worldwide (according to the World Health Organization’s statistics; 24 December 2023) and our knowledge concerning its possible course and accompanying late consequences is more and more detailed, we are still discovering new aspects of this phenomenon [3]. Initially, COVID-19 was perceived as a disease of the respiratory tract. Nonetheless, the list of possible extrapulmonary symptoms and complications became longer and longer [4,5,6,7,8]. Finally, the spectrum of COVID-19-related manifestations turned out to be associated with a broad range of systemic symptoms, e.g., neurological and dermatological syndromes, myocardial dysfunction, hyperglycemia, disorders of the gastrointestinal tract and acute kidney failure [9,10,11,12,13,14]. Concomitantly, a lot of observations also started to suggest an existing relationship between SARS-CoV-2 and chronic liver diseases (CLDs), indicating hepatic manifestations of the infection and the exacerbation of existing liver pathologies in the case of COVID-19 [15,16,17,18,19,20,21]. Further, pre-existing systemic disorders were also proved to modify the natural history of COVID-19, usually exacerbating its course and resulting in a more complex presentation of the infection. Special attention should be paid to patients with cardiovascular pathologies, diabetes and dyslipidemia. Insulin resistance and impaired immune response are usually enumerated in such circumstances as participating cofactors. Of note, an inflammatory background might impair the production of adiponectin within visceral adipose tissue, progressing underlying inflammation. These speculations have already been confirmed by numerous findings of conducted investigations showing that the population of COVID-19 patients with lower expression of adiponectin is more prone to developing respiratory failure [22]. A notable expression of a metalloproteinase, angiotensin-converting enzyme 2 (ACE2), a functional receptor for SARS-CoV-2, in the kidneys might be related to increased risk of acute tubular necrosis. Pre-existing pulmonary disabilities constitute another undeniable condition increasing a potential outcome of ongoing COVID-19. Simultaneously, *the gut–lung axis* reflects a bidirectional relationship between residents of the bacterial flora within the gastrointestinal and respiratory tracts. Thus, dysbiosis (quite often related to treatment with antibiotics and steroids) might alter the integrity of the intestinal barrier, resulting in an increased risk of secondary infections. Secondly, pulmonary-derived hypoxia can induce necrotic lesions in the cells of the gut. On the other hand, neurological manifestations in the early stages of COVID-19 may be perceived as indicators of poor clinical outcome. Furthermore, coexisting inflammation-induced hypercoagulability predisposes patients to developing strokes. Another crucial aspect of potential implications due to SARS-CoV-2 infection is related to possible autoimmune cross-reactions. As a result, idiopathic thrombocytopenic purpura or autoimmune hemolytic anemia might be developed [23]. A primary pulmonary disorder turned out to be a multiorgan complex pathology—this is the major lesson of the pandemic. Due to the notable involvement of the liver in the systemic manifestation of COVID-19, a lot of investigations were conducted to explore this relationship. Simultaneously, the context of alcohol dependence and the progression of ALD during the pandemic were observed. The aforementioned phenomena are essential from clinical and social perspectives. Because the data on these issues are not fully systemized, we decided to explore the available literature and present the current state of knowledge concerning COVID-19, alcohol abuse and possible hepatic complications in the most comprehensive way.

## 2. Hepatic Face of Novel Coronavirus Infection

The liver has been described as the second organ after the lungs to be involved in the course of the disease, resulting in hepatobiliary complications among up to 29% of patients [24,25,26,27]. Chen et al. found patients infected with coronavirus and coexisting abnormalities in liver tests to present a greater risk of systemic inflammatory response syndrome (SIRS) and higher overall mortality rates [28]. Another hepatic perspective related to COVID-19 is the risk of adverse events that may occur in the course of treatment; in the majority of cases, they are reflected by mild hypertransaminasemia [29]. In the scope of these speculations, we decided to present gathered data of patients suffering from alcohol-related liver disease (ALD) who were infected with SARS-CoV-2. The manifestation of infection by coronavirus mainly concerns symptoms related to the respiratory tract. However, in the course of the pandemic, a more and more complex nature of COVID-19 started to appear [30]. It turned out that a greater number of coexisting conditions among infected individuals (e.g., cardiovascular disorders, kidney failure, diabetes, cancer, obesity, neurodegenerative diseases and alcohol consumption) may predispose them to a more severe course of the infection [31,32,33,34,35]. From 2020–2022, at least 2336 manuscripts focused on coronavirus and its hepatic implications were published, and we are still discovering new aspects in this area [36]. Therefore, the involvement of the liver in the presentation of COVID-19 is still an issue of great importance, requiring further investigations. Simultaneously, the pandemic and lockdowns were related to the increased consumption of alcohol in society. This combination created the background to perceive ALD as a significant underlying factor in the natural history of coronavirus infection. Due to these considerations, we decided to gather already collected data on COVID-19, ethanol and ALD in a single manuscript.

## 3. SARS-CoV-2 and Liver—Direct or Indirect Implications?

Extrapulmonary manifestation of SARS-CoV-2 infection might even overtake pulmonary presentation [37,38,39]. Usually, the presentation with gastrointestinal symptoms involves the increased prevalence of hypertransaminasemia and liver injury. This is mainly due to the hepatic expression of metalloproteinase—angiotensin-converting enzyme 2 (ACE2)—a functional receptor for SARS-CoV-2 and its spike-I glycoprotein [40,41,42]. It was described in cholangiocytes and hepatocytes. During this initial stage of the infection, the renin–angiotensin system and peroxisome proliferator-activated receptor signaling pathway can be perceived as triggering factors. Thus, the liver cytopathic injury might be developed as a direct result of infection [27,43,44,45,46,47]. Simultaneously, a developing SIRS together with coexisting hepatic anoxia due to SARS-CoV-2-related respiratory failure can indirectly impair liver function The phenomenon of hepatic disorders in the overall picture of COVID-19 concerns from 2.5% up to 45.71% individuals and this wide range can be explained by the presence of diversified subpopulations among CLD patients and different cut-offs of liver test results applied in studies. In the majority of cases, hepatic manifestation of COVID-19 is asymptomatic and its only visible proof is elevation in liver enzymes. According to previous analyses, these disturbances concern mainly aminotransferases (approximately 20% of COVID-19 patients); about 15% of cases might present with an increased level of gamma-glutamyltransferase (GGT), 9.7% of cases with bilirubin and 4% of cases with alkaline phosphatase (ALP) [48,49,50]. The potential relationship between COVID-19 and liver disorders might be considered from at least two perspectives. Infection with SARS-CoV-2 can manifest with liver disorders or exacerbate already existing hepatic problems. Metabolic-associated fatty liver disease (MAFLD) turned out to be a triggering factor for COVID-19 [51,52]. Alterations in immune functions among individuals with liver steatosis are commonly seen and reflected by the increased concentration of interleukin (IL) 6 in their blood [53]. After the lungs, the liver is the second organ most frequently affected by COVID-19 [54]. Concurrently, individuals with chronic or acute liver disorders are prone to developing exacerbation due to coronavirus infection; this scheme was already confirmed [55]. It is assumed that about 3% of all COVID-19 patients suffer from underlying CLD [56]. Furthermore, in the case of autoimmune liver pathologies, the treatment based on immunosuppressants may constitute a triggering factor in liver injury after the development of infection with SARS-CoV-2 [57]. Regardless of the basic status of the patient (with or without previous liver failure), the management of COVID-19 patients should routinely include the assessment of liver function tests (LFTs).

## 4. Certain Molecular Pathways in COVID-19-Induced Liver Failure

Regardless of the nature of liver failure (direct or indirect) due to COVID-19, several possible hypotheses are believed to be linked with hepatic injury in such circumstances. Among others, the following are worth emphasizing: the direct viral effect, drug hepatotoxicity, systemic inflammatory response (cytokine storm), decompensation of pre-existing liver disease and hypoxic liver injury [58,59,60,61,62]. No pathognomonic features of hepatitis due to SARS-CoV-2 infection have been found in liver biopsy specimens. Nonetheless, some data revealed the presence of tropism of the virus in the liver [63,64,65]. This organ can constitute a potential target for infection because of the presence of the SARS-CoV-2 receptor, ACE2, on the surface of the hepatic Kupffer cells, hepatocytes and cholangiocytes [66,67]. Moreover, SARS-CoV-2 spike structures were isolated from the cytoplasm of hepatocytes in COVID-19 patients [68]. The replication of the virus and direct invasion of liver cells were also confirmed by the spatial presence of SARS-CoV-2 RNA and spike protein in hepatocytes [69]. On the other hand, some reports indicated a lower expression of the coronavirus in the liver compared to the lungs, suggesting the lack of certain cytopathic abnormalities in hepatocytes during COVID-19 presentation [70]. Therefore, it is impossible to describe the involvement of the liver in coronavirus infection as a rule. Nevertheless, available data suggest the possibility of such a scenario. Table 1 presents the most significant relationships between COVID-19 and acute liver failure published in the literature.

### 4.1. Inflammatory Storm

Multiple inflammatory particles are secreted in the course of COVID-19 as the response of the innate immune system. A key role is played by: tumor necrosis factor alpha (TNF-α), pro-inflammatory T helper 17 cells, together with IL-2, IL-6, IL-7, IL-8, IL-10 and IL-1B. This cytokine storm is reflected by the recruitment of macrophages and histopathological findings within hepatocytes, proving their steatosis, lymphocyte infiltration and Kupffer cells’ hyperplasia [91,92,93,94,95]. Further, this inflammatory response may result in further pathological events, including circulatory disorders, thrombotic complications and multiorgan failure [57,96,97,98,99,100,101]. Finally, this inflammatory hurricane spreads through the whole body and it becomes nearly impossible to decide whether the liver was affected as a primary or secondary organ.

### 4.2. Hypoxic–Ischemic Liver Failure

Hypoxic harm is another underlying problem in COVID-19 patients, which can eventually promote hepatic dysfunction. Firstly, shock might develop as a consequence of infection with the coronavirus and consequently result in reduced liver perfusion. Simultaneously, coexisting respiratory failure is another underlying factor involving hypoxic trauma to the liver independently of ischemia [102,103,104,105,106]. Finally, the implication of COVID-19 in the phenomena of microangiopathy and thromboembolism interacts with impaired liver blood supply in such circumstances. Of note, even hypertransaminasemia was found to correlate with alterations in coagulopathy markers (i.e., prothrombin time, international normalized ratio, fibrinogen, D-dimer, fibrin) in individuals infected with SARS-CoV-2. In addition, an increased metabolic activity of the liver in the situation of cardiac, vascular and pulmonary involvement due to COVID-19 constitutes a different potential background for its hypoxic–ischemic harm [107,108,109,110,111]. Histopathological evaluation in the autopsies of COVID-19 patients revealed the presence of features characteristic of hypoxic–ischemic liver damage (i.e., ischemic-type hepatic necrosis and lipid droplet accumulation). A notable hypertransaminasemia should accompany morphological changes within hepatocytes in order to fulfill criteria of hypoxic hepatitis; nonetheless, it is not a typical pattern for COVID-19, according to already gathered data. Because of these discrepancies, hypoxia–reperfusion is postulated to be another underlying mechanism. Blood congestion in the liver results from impaired hepatic venous drainage and it was visualized in more than one third of histopathological surveys [112,113,114,115]. From a clinical point of view, causative factors responsible for the development of hepatic congestion in the course of coronavirus infection are: right-sided heart failure, decompensation of existing heart failure and pulmonary thromboembolism. Of note, a correlation between the dysfunction of the liver and mechanical ventilation introduced in COVID-19 patients was observed. It mainly concerned conditions with high rates of positive end-expiratory pressure [95,116,117]. In everyday clinical practice, it is advisable to pay special attention to LFTs in patients requiring mechanical ventilation.

### 4.3. Impaired Iron Metabolism

A disrupted iron metabolism might be considered as an accompanying mechanism that contributes to liver damage. SARS-CoV-2 is believed to recognize serum iron-binding particles in various organs (including the liver) and to cause the direct influx of iron ions to cells. Simultaneously, according to the exacerbated inflammatory response, IL-6 promotes the synthesis of hepcidin within hepatocytes, resulting in the sequestration of iron in cells. This double mechanism, supported by the generation of reactive oxygen species and active lipids, finally leads to ferroptosis—a novel form of cell death. Even though this phenomenon requires further investigations, it can be hypothesized as a potential target for treatment in COVID-19 [118,119,120,121,122,123].

### 4.4. Side Effects of Underlying Treatment

Another underlying issue is the treatment of COVID-19 and possible related and unexpected side effects involving hepatotoxic consequences. They might be associated with various medications: antivirals (e.g., remdesivir, lopinavir/ritonavir, favipiravir), hydroxychloroquine, antibiotics (e.g. azithromycin), corticosteroids, tocilizumab and antipyretics, mainly acetaminophen [124,125]. The data already collected in this field indicated quite a mild course of liver injury during the therapy with remdesivir, usually reflected by transient increases in liver enzymes [126,127,128,129]. Analysis of a several COVID-19 patients treated with fanipiravir revealed its potential hepatotoxic effect, too. Nevertheless, a systematic review on this agent did not support the association. A more significant role might be played here by lopinavir and ritonavir. They were found to increase the risk of liver injury in monotherapy or in combination with ribavirin and interferon-β [86,130,131]. The use of corticosteroids in COVID-19 patients, especially among individuals demanding oxygen supplement therapy, constitutes another issue. Some data collected from case reports indicate a possibility of steroid-induced hypertransaminasemia. Simultaneously, they are also proposed in patients infected with the coronavirus as agents helpful in the treatment of drug-induced liver injury. A potential explanation for liver toxicity linked to corticosteroids may be their contribution to the development of non-alcoholic steatohepatitis (NASH) [132,133]. Tocilizumab, acting against IL-6, is also worth mentioning. Some reports indicate its involvement in mild hepatitis in subjects with COVID-19; nonetheless, acute liver failure is a rare possible occurrence [134,135]. Commonly used in the treatment of malaria, hydroxychloroquine and chloroquine were also applied as suggested drugs in patients infected with SARS-CoV-2. In this case, only single reports revealed a potential impairment of liver function [136,137,138]. As for antibiotics included in the list of agents used in the course of COVID-19, azithromycin was observed to provoke hepatocellular and cholestatic injury to the liver [139,140,141]. Further, in the field of antiparasitic drugs, ivermectin seems to be related to an episode of severe hepatitis in patients with coronavirus infection. However, the data on this relationship are too weak and further investigations to generalize this association are required [142]. Finally, taking into consideration a causative role of acetaminophen in the course of liver dysfunction in COVID-19 patients, it is not an established pattern, probably due to possible hepatic side effects mainly in the case of its overdosing [143,144]. Of note, it should be emphasized that each combination of different agents administered to individuals with COVID-19 might constitute the background of unexpected impairments in liver function because of their synergistic effect. We are still discovering new possible dependences in this area [145]. Nevertheless, the potential development of side effects related to conducted therapies is unpredictable. Therefore, careful monitoring of LFTs seems to be the best approach.

## 5. COVID-19 in Patients without Previously Diagnosed Liver Disorders—Should We Expect Hepatic Complications?

The last four years have given us a background to form some possible relationships between specific risk factors predisposing patients without coexisting liver failure to developing hepatic abnormalities after being infected with SARS-CoV-2 [146,147,148]. Among others, there are some basic parameters in laboratory findings, e.g., marked inflammation reflected by the elevation in C-reactive protein (CRP) and erythrocyte sedimentation rate (ESR) or lymphopenia. Generally speaking, SIRS and cytokine storm observed in the course of coronavirus infection are perceived as direct triggering factors leading to the impairment of liver function [77,80,149,150]. Epigenetic features (obesity, advanced age and female gender) constitute underlying associated conditions. Significantly greater risk of hepatic complications in men is usually explained by more pronounced liver steatosis and alcohol intake. Previous observations revealed that enzymatic features of liver disorder accompanying COVID-19 might predict a more severe course of the disease, poorer overall outcome and a higher rate of mortality [151,152,153,154,155]. In general, individuals affected with SARS-CoV-2 infection usually present a hepatocellular pattern of liver injury (with a threefold increase in aminotransferases above the upper limit of normal (ULN)) or a mixed one (hepatocellular pattern together with a twofold increase in ALP/GGT) [156]. Fulfilling the above-mentioned criteria indicated a significant risk of more severe presentation due to COVID-19. Moreover, hypoalbuminemia, higher levels of alkaline transferase (ALT) and elevated values of ferritin and IL-6 constituted other prognostics of liver failure in healthy patients affected with SARS-CoV-2 infection [157,158]. It can be assumed that the liver becomes an organ transducing inflammatory signals during the disease.

### 5.1. Hepatic Imaging Findings in COVID-19 Patients

Some notable findings in imaging studies of the liver in the course of COVID-19 have also been reported. They concerned hepatic hypodensity together with fat stranding around the gallbladder in the majority of cases. Of note, lower values of liver CT attenuation and liver-to-spleen attenuation ratio were shown to correlate with the severity of COVID-19. Simultaneously, the presence of liver steatosis might concern up to 11% of cases, increasing the risk of hepatic injury [159,160]. Even though the findings of abdominal ultrasound examination are usually irrelevant in patients with normal results of liver function tests, in the case of significant increase in aminotransferases, vascular disorders and cholestatic abnormalities can be observed and related to a higher rate of mortality [161,162]. The data on post-mortem assessment of nineteen COVID-19 individuals without a previous known history concerning liver disorders revealed the presence of hepatic abnormalities in seventeen of them (89%) [163]. According to available literature, it is hard to generalize the main type of cytopathic effect of SARS-CoV-2 exerted on the liver. However, it appears to be evident that such abnormalities could be perceived in the future as markers of the severity of the disease.

### 5.2. The Issue of Vaccination against COVID-19

The use of vaccination against COVID-19 constitutes another issue worth mentioning in the scope of potential liver failure. According to already gathered data, it can be assumed that the majority of affected individuals develop a hepatocellular form of injury. It is suggested that a possible background of such a reaction might be related to the similarity between S protein (encoded by the vaccines) and liver-specific proteins [164,165,166,167]. In consequence, autoimmune-like hepatitis might be developed. Histopathological assessments of liver biopsy specimens performed in patients with this complication revealed portal lymphoplasmacytic infiltration with interface hepatitis. In clinical cases, it appears that the most appropriate and successful treatment in such occurrences is based on corticosteroids, with mortality of approximately 4% [168,169,170,171]. According to a recent systematic review, Moderna mRNA–1273 was described as the most significant vaccination related to immune-mediated liver injury, along with the following: Pfizer-BioNTech BNT162b2 mRNA and AstraZeneca ChAdOx1 nCoV-19 vaccine. Simultaneously, there is a growing body of evidence that inactivated vaccines might also exert hepatotoxic activity. Nevertheless, this issue requires further investigations. Clinical manifestation of liver injury was usually reflected by jaundice and elevation in values of: ALT, ALP and bilirubin [170,172,173]. Which diagnostic criteria should be fulfilled to identify anti-SARS-CoV-2 vaccine-related liver failure have been discussed. The underlying issue is the diagnostic strategy in such circumstances and especially whom it should mainly concern.

## 6. COVID-19 in Patients with Pre-Existing Liver Failure—General Observations

The perspective of four years of the pandemic due to coronavirus infection gave us a deep insight into its course and possible implications. Undoubtedly, we are still facing new findings in this field. Nevertheless, with the rapidly increasing number of new COVID-19 cases, our knowledge became more detailed—even in the area of hepatology. Because of the significant prevalence of CLD worldwide (112 million people according to the Global Burden of Disease (GBD) Study from 2017), the observation of the direct influence of coronavirus infection on its course is of crucial importance [174,175,176,177,178]. According to available data, it can be assumed that liver involvement is not a typical occurrence in COVID-19 patients and if it develops, usually, the course of liver injury is mild and transient, with no need for pharmacological treatment. Nonetheless, the situation becomes more complicated among individuals with pre-existing hepatic disorders—mainly non-alcoholic fatty liver disease (NAFLD), MAFLD and cirrhosis [179,180,181]. On the other hand, previously diagnosed viral hepatitis was not found as a triggering factor for liver damage after being infected with the coronavirus [182]. According to global data, the overall risk of a severe course and death because of COVID-19 in patients with CLD becomes 2.44 times greater in comparison to the population with no hepatic problems [183,184,185]. The global multicenter experience (the United States, Europe and China) with cirrhotic patients treated due to SARS-CoV-2 infection gives the perspective of the significantly higher mortality rate among these individuals (up to 35%) compared to the non-cirrhotic population [186,187,188]. According to the latest analyses of data, even though the results of observations seem to be conflicting, cirrhosis is predominantly described as an independent factor increasing mortality due to coronavirus infection [189]. Therefore, the basic status of liver function related to infection with SARS-CoV-2 should be perceived as an important parameter in the case of COVID-19 [190,191]. How about the exact explanation of the notable involvement of cirrhosis in the presentation of COVID-19? The most probable scenario concerns immunological disturbances observed in cirrhotic patients. Dysregulation of innate and adaptive immunity may be followed by the greater susceptibility to developing ARDS and SIRS—phenomena which constitute natural sequelae of coronavirus infection. Moving further, cirrhotic patients are characterized by: intestinal dysbiosis, altered lymphocytes together with neutrophil function, Toll-like receptor upregulation, impaired macrophage activation and decreased expression of the complement system [192,193,194,195,196,197]. Figure 1 presents cirrhosis-related immunological disorders triggering the course of COVID-19.

## 7. Ethanol, Liver and COVID-19—A Global Perspective

One could ask if there are any specific patterns of COVID-19 in ALD patients. And consequently, is this population worth exploring regarding this problem? Put briefly, it seems so. Because of its worldwide significance and potential complications, ALD is often considered as a condition that might be exacerbated in the course of various newly developed medical conditions [198]. Infection with SARS-CoV-2 is the ideal example of such occurrences [199,200,201]. Of note, according to a meta-analysis, since the beginning of COVID-19 pandemic, ALD has been indicated as the most common reason for listing for and the fastest increasing cause of liver transplantation (LT) [202]. Another source of data revealed that in 2020 ALD constituted 40.1% of all LT listings and this result was greater than cases related to hepatitis C virus (HCV) and NASH taken together. Another retrospective study performed in the United States between March 2020 and September 2022 showed a more pronounced mortality due to ALD, cirrhosis and hepatic failure in all; the excess risks were 1.4–2.8 times higher [182,203]. One could ask: why? Everything started in the local lives of societies. The intake of alcohol, especially during the first phase of the pandemic, turned out to be an extremely significant medical problem, with more marked harmful alcohol consumption, and this tendency was similar in different parts of the world compared to the previous year [204,205,206]. This phenomenon from everyday life was reflected by the increasing number of ALD cases together with its more severe course all over the world (e.g., in the United States, Great Britain and China) [207]. What appeared to be especially alarming, according to available data, was that the most significant group requiring LT were young adults [208,209,210,211]. Fortunately, despite the pandemic era and a greater number of conducted transplants, 3-month post-LT recipient survival time remained at a similar level of above 96% [212].

### Coronavirus in ALD Patients

The definite majority of already obtained data concerning the outcome of patients with COVID-19 and underlying liver disorders focus on liver steatosis. Previous observations indicated that BMI and the presence of MAFLD might constitute risk factors for a more severe course of COVID-19 [213]. The number of reports on patients with ALD infected with SARS-CoV-2 seems to be somewhat limited. Causative factors might be related to social dependencies and a generally lower socioeconomic status of patients with alcohol abuse together with their worse physical condition due to portal hypertension and metabolic liver failure at baseline. Thus, coexisting COVID-19 does not have to be perceived as an accompanying decompensating factor in all cases because of the nature of ALD. Therefore, a potential significant role of ALD in the course of SARS-CoV-2 may be mistakenly overlooked [214,215,216,217]. Another fundamental issue is that a potential influence of alcohol use disorder (AUD) on the pattern of liver injury in the case of underlying COVID-19 and possible differences in hepatic manifestation of the disease regarding a previous alcohol abuse history or its absence must be clarified. The inflammatory background of these two conditions appears to be of crucial importance in relation to searching for potential dependencies, even though it can only be speculated which factor is this triggering one: ALD or coronavirus infection [218,219,220]. The data concerning this aspect are rather scarce, nevertheless some attempts to identify the alcoholic background of hepatic complications due to infection with SARS-CoV-2 have already been made. Relating to a former pre-existing liver disorder among observed COVID-19 patients, excessive alcohol intake was associated with a 5.5-fold greater risk of intensive care during hospitalization and 10 times increased probability of developing liver failure [202,221,222]. Another retrospective analysis performed in the United States in 2002 revealed a general decrease in gastrointestinal diagnoses with a notable exception of ALD (which increased by 7.8%) [223]. Sobotka et al. decided to compare the outcome of patients hospitalized due to alcoholic hepatitis and alcohol-related liver cirrhosis (ALC) in two different periods of time: during the COVID-19 pandemic and in the pre-pandemic setting. The major conclusion was a definitely worse prognosis in ALD patients during the pandemic. In the course of alcohol hepatitis in the pandemic, complications like hepatic encephalopathy and variceal hemorrhage were more common; similarly, patients more often required oxygen, vasopressors and hemodialysis. On the other hand, cirrhotic patients observed during the COVID-19 pandemic compared to pre-pandemic conditions were found to have higher results in MELD-Na score and were at a higher risk of developing hepatic encephalopathy, ascites and spontaneous bacterial peritonitis; inpatient mortality was also higher. It is hard to decide whether the reason for the above-mentioned observations was directly related to the greater consumption of alcohol in the pandemic period, nonetheless it can be perceived as a probable reason for these differences [224]. A similar observation in a larger group of patients showed an increased number of hospitalizations among patients with AH and ALD during the COVID-19 pandemic. Furthermore, the mortality in the course of ALD compared to pre-pandemic years was greater, too [225,226].

## 8. ALC and SARS-CoV-2—Any Specific Patterns and Relationships?

Nuovo et al. tried to find if liver biopsy specimens from patients who died of COVID-19 differ significantly regarding their previous medical history: AUD, NASH or no previously diagnosed liver failure. Firstly, it turned out that individuals with alcohol abuse presented a notably greater population of SARS-CoV-2-spike-protein-positive cells in comparison to the other people included in the survey. Nonetheless, nucleocapsid protein was extracted only from lung samples; it was undetectable in liver biopsies. On the other hand, liver extracts obtained from the AUD group had a significantly greater content of detectable viral spike and matrix proteins of SARS-CoV-2. Moving further, the density of viral spike protein in AUD/COVID-19 livers compared to NASH and non-NASH biopsy specimens was definitely more pronounced (*p* < 0.001). The study also revealed a 10-fold greater number of ACE2-positive cells (represented mainly by activated hepatic stellate cells) within the liver among persons with known AUD. This finding was confirmed by an accompanying more pronounced density of liver cells that endocytosed SARS-CoV-2 protein in this study group [227]. Despite valuable results, the number of patients qualified for this investigation is undeniably very small. The basic issue is the probable background of possible differences in the hepatic manifestation of COVID-19 between cases with ALD and other liver pathologies. The explanation could be related to inflammation, crucial in the course of alcoholic liver failure as well as in the natural history of coronavirus infection. To date, the number of investigations dedicated to this problem is highly limited, so further explorations are required [228,229]. Nevertheless, the analysis of individuals with non-cirrhotic CLD indicated ALD as the state predisposing to increased mortality [230,231]. Another observation of 867 CLD patients with COVID-19 showed ALD, decompensated cirrhosis and HCC as liver-specific factors responsible for higher overall mortality [232]. Corresponding data were collected by Marjot et al. after creating the international registry of cirrhotic (*n* = 386) and non-cirrhotic (*n* = 359) patients with CLD, additionally affected with SARS-CoV-2 infection [233]. It presented that liver fibrosis constitutes an essential factor worsening general prognosis; the mortality rate in two observed groups was 32% and 8%, respectively. Of note, mortality was especially pronounced among individuals with more advanced cirrhosis and with an alcoholic background of liver disease. On the other hand, NAFLD and hepatitis B virus (HBV) were found to correlate negatively with mortality. This seems to be the first international study showing a direct link between ALD and poor prognosis of COVID-19 in CLD patients; ALD increased the risk of death 1.8-fold. Additionally, after a separate examination of the cirrhotic subgroup, no other factor related to the etiology of disease was found to be significant. According to the above-mentioned survey, it should only 6% of the ALD subpopulation were individuals without liver cirrhosis. Thus, the stage of CLD was already significantly advanced. Nonetheless, it is quite a common feature that patients with ALD are rarely diagnosed at an early phase of this pathology [234,235,236]. Of note, ALD was identified as a risk factor of poor COVID-19 outcome in another survey on 80 subjects with CLD, too [237]. 

### 8.1. Alcohol-Mediated Pathological Pathways and Coronavirus—A Complex Scenario

On the basis of current knowledge, it is extremely important to speculate about the role of ALD as an independent factor worsening the general outcome of CLD patients with coronavirus infection. However, it can be generally assumed that excessive alcohol intake exerts various immunomodulating effects, beyond hepatotoxic consequences. Therefore, the increased alcohol-induced probability of developing bacterial or viral infections together with ARDS in sepsis is well-known [238,239]. Simultaneously, already gathered data indicate that alcohol abuse might worsen the course of COVID-19. The most probable underlying mechanisms include: an increased risk of developing ARDS in alcohol addicts and a greater number of comorbidities among ALD individuals. Figure 2 shows mechanisms related to ethanol that participate in the presentation of COVID-19.

Malnutrition and metabolic syndrome constitute other underlying issues [240,241]. The development of leaky gut and microbiota dysbiosis can be perceived as cofactors [242,243]. Alcohol together with COVID-19 modifies the content of microbiota and alters the permeability of intestinal cell junctions. In consequence, translocation of lipopolysaccharide (LPS) and pathogen-associated molecular patterns from the gut occurs, allowing them to bind to pattern recognition receptors found within, e.g., macrophages and hepatic Kupffer cells. This cascade exacerbates the release of proinflammatory mediators [244,245]. Ethanol induces the release of reactive oxygen species and promotes the secretion of various proinflammatory chemokines, simultaneously inhibiting the production of anti-inflammatory particles. Finally, the integrity of the epithelium becomes disrupted, leading to alveolar epithelial dysfunction and decreased concentration of pulmonary antioxidants [246]. Additionally, in previous studies, alveolar macrophages obtained from animal models chronically drinking ethanol presented intensified inflammatory response to LPS and respiratory syncytial virus [247,248]. Except for pulmonary macrophages, the populations within the myocardium and central nervous system are affected, too. Moving further, ethanol leads to the suppression and dysfunction of T lymphocytes, resulting in altered adaptive immunity. The phenomenon reflects the immune imbalance typical of coronavirus infection. Additionally, chronic alcohol intake might lower antibody response with vaccinations against SARS-CoV-2 [249]. Heavy alcohol consumption together with, e.g., a lack of physical activity and cigarette use was included in the list of modifiable lifestyle factors directly influencing the potential outcome of COVID-19 [250]. Moreover, following a bland diet was described as another factor increasing the susceptibility of being infected with the coronavirus [251]. However, it still remains unclear how ALD might specifically affect the course of the disease [252,253].

### 8.2. Characteristic Features of ALD Patients in the Scope of COVID-19

The treatment of patients with alcoholic hepatitis with steroids should also be emphasized. Nevertheless, its overall role in the final outcome of the course of COVID-19 has not been established yet. Previous studies showed that such management might increase the risk of bacterial infections and hypokalemia, but the direct relationship with increased mortality due to coronavirus infection was not proved [254,255]. More than 430,000 volunteers with COVID-19 within another study presented their drinking alcohol-drinking habits. The results of this self-reported ethanol consumption revealed no relationship between heavy drinking and increased mortality risk of COVID-19 or other infections; moderate drinkers presented a lower risk. Finally, former drinkers were at risk of higher mortality due to infectious diseases, but not COVID-19 [256]. Therefore, the data concerning this issue sound somewhat conflicting, especially from the social perspective [257]. The study of 1325 patients with ALC and confirmed COVID-19 compared to 1135 matched infected individuals without liver disease did not show ALC to be a factor increasing mortality in the course of SARS-CoV-2 infection. Nevertheless, cirrhotic patients presented a notably greater number of underlying complications (e.g., septic shock, upper extremity venous thromboembolism or atrial fibrillation) [258,259]. Other researchers found a positive association between decompensation of CLD and in-hospital mortality in COVID-19 patients. Moving further, ALD constituted a risk factor for the need for introducing mechanical ventilation in the course of coronavirus infection [260,261]. Corresponding data were obtained by Wang et al.—ALD and ALC were notably related to a more severe course of COVID-19 [262]. Interesting observations were made by Huang et al. in a meta-analysis on systemic disturbances due to exposure to ethanol among patients with an identified coronavirus. Surprisingly, alcohol was the factor provoking the stimulation of the hepatic fibrosis signaling pathway directly by SARS-CoV-2 [263].

## 9. Conclusions

Even though our knowledge regarding possible hepatic complications in the course of COVID-19 had become more comprehensive, the management of such patients still appears to be challenging. As well as worldwide experience based on particular CLD cases with coexisting coronavirus infection, the guidance of the European Association for the Study of the Liver (EASL) can be perceived as a helpful tool in this area. In this summary, ALD was not described as a factor increasing the risk of developing SARS-CoV-2 infection, but a predictor of potentially greater mortality in the course of the disease compared to other backgrounds of CLD [264]. We still need more surveys to be performed in this field, nevertheless, the relationship between ALD and COVID-19 is undeniable and crucial from a clinical perspective. Both alcohol abuse and coronavirus infection, together with the involvement of hepatic complications in the course of these pathologies, may be perceived as self-perpetuating mechanisms. The major aims of this review were to: emphasize complex relationships between these phenomena and highlight their clinical significance. Taking care of individuals with ALD additionally infected with SARS-CoV-2 should be based on a comprehensive attitude including an anti-inflammatory strategy and the support of liver function.

## Figures and Tables

**Figure 1 ijms-25-02212-f001:**
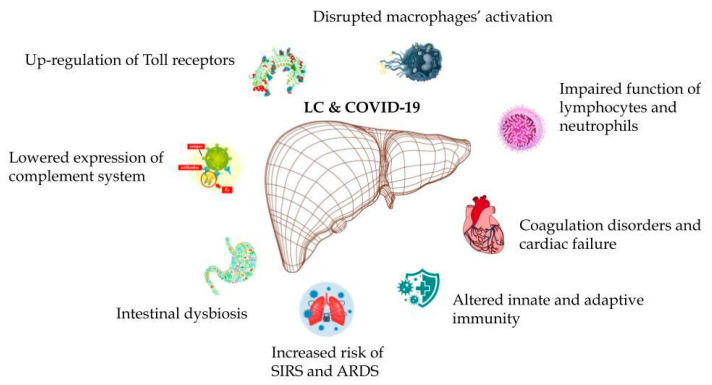
Mechanisms of liver cirrhosis-related disorders involved in the presentation of COVID-19. LC—liver cirrhosis, SIRS—systemic inflammatory response syndrome, ARDS—acute respiratory distress syndrome.

**Figure 2 ijms-25-02212-f002:**
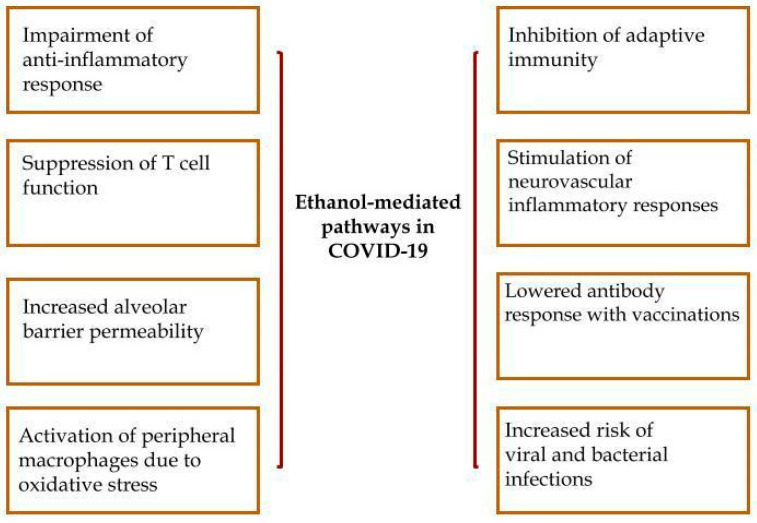
Ethanol-induced injury in the course of SARS-CoV-2 infection.

**Table 1 ijms-25-02212-t001:** Acute liver injury in the course of COVID-19.

	Research Type	Markers of ALF (ALT, AST or TBIL)	Cases	ALF	Prognostic Indicator
1. Bogline 2022 [71]	Retrospective	>1 ULN	434	123	NR
2. Desai 2020 [72]	Retrospective	>1 ULN	639	476	b
3. Shousha 2021 [73]	Cohort study	>1 ULN	428	137	b
4. Wang 2020 [74]	Cohort study	>1 ULN	657	303	a
5. Wang 2020 [75]	Cohort study	>1 ULN	339	96	b
6. Piano 2020 [76]	Retrospective	>1 ULN	565	329	bc
7. Phipps 2020 [77]	Retrospective	>2 ULN	2273	489	ab
8. Mishra 2021 [78]	Retrospective	>1 ULN	348	184	c
9. Chew 2021 [79]	Cohort study	>5 ULN	834	105	bc
10. Chen 2021 [80]	Cohort study	>1 ULN	830	227	ab
11. Krishnan 2022 [81]	Retrospective	>1 ULN	3830	2698	ab
12. Zhang 2021 [82]	Retrospective	>1 ULN	440	254	abc
13. BJ 2021 [83]	Retrospective	>1 ULN	382	159	abc
14. Chu 2020 [84]	Cohort study	>2 ULN	838	429	ab
15. Hassanin 2021 [85]	Retrospective	>2 ULN	1238	296	c
16. Cai 2020 [86]	Cohort study	>1 ULN	417	192	a
17. Fu 2020 [87]	Retrospective	>1 ULN	482	142	abc
18. Siddiqui 2021 [88]	Retrospective	>4 ULN	1935	396	NR
19. Faghih 2022 [89]	Cohort study	>1 ULN	1017	324	ab
20. Cholongitas 2022 [90]	Retrospective	>5 ULN	1046	53	c

a—non-severe vs. severe, b—survival vs. death, c—hospital stays, NR—not reported, ALF—acute liver failure, ALT—alanine aminotransferase, AST—aspartate transaminase, TBIL—total bilirubin abnormal, ULN—upper limit of normal.

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
