# Peer review of "COVID-19, Possible Hepatic Pathways and Alcohol Abuse—What Do We Know up to 2023?"

_ijms, 2024, doi:10.3390/ijms25042212_

Round 1
Reviewer 1 Report
Comments and Suggestions for Authors
This article highlights the correlation between COVID-19 and various liver processes. The topic is relevant, but some shortcomings identified in both content and form need to be addressed based on the specific recommendations below:
0. The title need not be fully capitalised.
1. The abstract should be presented in a more formal way as it seems much too informal (as in an oral presentation) for a scientific paper.
2. Abbreviations should be treated separately in the abstract from the main text. Abbreviations are mentioned and explained at the beginning once, then only abbreviated forms are used. If only one abbreviation is used, no abbreviation is necessary (e.g. SARS-CoV-2 in the abstract, etc.). Check and correct abbreviations throughout the manuscript.
3. [1], [2] according to the rules of bibliography insertion should be [1,2], and [3]-[7] should be [3-7]. Check and correct throughout the manuscript.
4. The introduction is poorly presented in relation to the complexity of the topic. First, it consists of one large paragraph which is very difficult to follow. I recommend dividing the information into smaller and clearer paragraphs (do this with the whole manuscript). Then, it discusses too briefly or not at all in some cases essential topics in the introduction of a topic dealing with COVID-19 (therapy, associated disease, especially as it refers to liver pathology, comorbidities, biochemical processes and biomarkers). I recommend and suggest including current and relevant publications in the more detailed construction of the introduction of this manuscript. I suggest referring to: PMID: 36406478; PMID: 35131656; PMID: 36211634.
5. The present manuscript lacks in the last part of the introduction the purpose of the paper which should be clearly stated, insisting on the contribution this manuscript can bring to the scientific literature and the novelty, especially as many publications address the topic COVID-19-associated pathologies and extrapulmonary implications.
6. Section two (representing an entire chapter) is only half a page. It is advisable to detail this part, especially since the entire manuscript is only 10 pages long, which may be considered slightly too little for a comprehensive review.
7. A detailed table with data from clinical trials / important findings from systematic reviews / meta-analyses / observational or cohort studies that have targeted liver pathology in COVID-19 patients is essential for clarity.
8. Improve the conclusions by addressing them also under the aspect of future research directions and the references should follow the same formatting as the text itself (palatino linotype of 9 for references; DOI code is not mandatory).
Author Response
Thank you for your valuable comments that improved the manuscript. We tried to follow all the suggestions. Changes within the article were written in the purple colour.
This article highlights the correlation between COVID-19 and various liver processes. The topic is relevant, but some shortcomings identified in both content and form need to be addressed based on the specific recommendations below:
- The title need not be fully capitalised.
The style of the title was modified.
- The abstract should be presented in a more formal way as it seems much too informal (as in an oral presentation) for a scientific paper.
The form of the abstract was improved in order to make its character more scientific.
- Abbreviations should be treated separately in the abstract from the main text. Abbreviations are mentioned and explained at the beginning once, then only abbreviated forms are used. If only one abbreviation is used, no abbreviation is necessary (e.g. SARS-CoV-2 in the abstract, etc.). Check and correct abbreviations throughout the manuscript.
The indicated corrections were introduced.
- [1], [2] according to the rules of bibliography insertion should be [1,2], and [3]-[7] should be [3-7]. Check and correct throughout the manuscript.
The style of bibliography was improved according to the suggestions.
- The introduction is poorly presented in relation to the complexity of the topic. First, it consists of one large paragraph which is very difficult to follow. I recommend dividing the information into smaller and clearer paragraphs (do this with the whole manuscript). Then, it discusses too briefly or not at all in some cases essential topics in the introduction of a topic dealing with COVID-19 (therapy, associated disease, especially as it refers to liver pathology, comorbidities, biochemical processes and biomarkers). I recommend and suggest including current and relevant publications in the more detailed construction of the introduction of this manuscript. I suggest referring to: PMID: 36406478; PMID: 35131656; PMID: 36211634.
The introduction was divided into subsections. The division of the rest of the manuscript was performed, too. Suggested articles were included in the bibliography [PMID: 36406478 (22), PMID: 35131656 (28), PMID: 36211634 (21)].
- The present manuscript lacks in the last part of the introduction the purpose of the paper which should be clearly stated, insisting on the contribution this manuscript can bring to the scientific literature and the novelty, especially as many publications address the topic COVID-19-associated pathologies and extrapulmonary implications.
The scientific soundness of the introduction was strengthened. We wrote about the context of the literature.
- Section two (representing an entire chapter) is only half a page. It is advisable to detail this part, especially since the entire manuscript is only 10 pages long, which may be considered slightly too little for a comprehensive review.
- A detailed table with data from clinical trials / important findings from systematic reviews / meta-analyses / observational or cohort studies that have targeted liver pathology in COVID-19 patients is essential for clarity.
The table (Table one) was placed in the main body of the manuscript.
- Improve the conclusions by addressing them also under the aspect of future research directions and the references should follow the same formatting as the text itself (palatino linotype of 9 for references; DOI code is not mandatory).
Conclusions and references were improved.
Reviewer 2 Report
Comments and Suggestions for Authors
This manuscript reviews the implications of COVID disease for the liver, which is a not-so-often discussed topic, since COVID has been mainly linked to respiratory disease and also, in ocassions, to diet and obesity and also social and psychological consequences of the pandemics and the lockdown.
Therefore, it is very interesting to find a review that fills in the gap in the field by focusing on the liver and reviews all the relevant existing literature linking COVID and liver.
I would suggest the authors to make more attractive and relevant Figures, by depicting the different cell types involved in the effects of the virus on the liver, and not only sentences with arrows.
Since the authors may be experts on the field of liver pathology I would also suggest to have a sentence at the end of each section stating a conclusion and/or take-home message.
Comments on the Quality of English LanguageNot only English language but also the style of writing should be carefully reviewed. For example, why is the title in capital letters? Why the first sentence of the Introduction states that COVID is an "idea"
Author Response
Thank you for your valuable comments that improved the manuscript. We tried to follow all the suggestions. Changes within the article were written in the purple colour.
This manuscript reviews the implications of COVID disease for the liver, which is a not-so-often discussed topic, since COVID has been mainly linked to respiratory disease and also, in ocassions, to diet and obesity and also social and psychological consequences of the pandemics and the lockdown.
Therefore, it is very interesting to find a review that fills in the gap in the field by focusing on the liver and reviews all the relevant existing literature linking COVID and liver.
I would suggest the authors to make more attractive and relevant Figures, by depicting the different cell types involved in the effects of the virus on the liver, and not only sentences with arrows.
Figure 1 was improved.
Since the authors may be experts on the field of liver pathology I would also suggest to have a sentence at the end of each section stating a conclusion and/or take-home message.
The majority of sections were supplied in conclusions.
Comments on the Quality of English Language
Not only English language but also the style of writing should be carefully reviewed. For example, why is the title in capital letters? Why the first sentence of the Introduction states that COVID is an "idea"
Vocabulary and grammatical constructions were checked by a qualified native speaker.

Round 2
Reviewer 1 Report
Comments and Suggestions for Authors
The manuscript has been significantly improved based on the suggestions received. A minor correction should be made before publication:
The first section should be named 'Introduction' according to the template provided by the journal, and the last paragraph of the introduction should contain the aim of the paper, coupled with the contribution to the literature and the novelty/research gap filled.
Author Response
Thank you for the comments. We followed your suggestions and improved the introduction according to them.